Fairness modeling for topics with different scales in short texts

Zhu Chuangying 1
Liang Yongyu 1 yogurtlcy@yeah.net
Liang Xinyuan 1
Zhong Limiao 1
Xie Fei 2
1 Guangxi Key Laboratory of Trusted Software, Guilin University of Electronic Technology , Guilin , China
2 Academic Affair Office, Guilin University of Electronic Technology , Guilin , China
Agrawal Rajeev
Electronic publication date: 2025 Jul 23
Publication date: 2025
Volume: 11
Electronic Location ID: e2936
Received 2025 Feb 3; Accepted 2025 May 12
Copyright: © 2025 Zhu et al.
Copyright year: 2025
Copyright holder: Zhu et al.
License: This is an open access article distributed under the terms of the Creative Commons Attribution License, which permits unrestricted use, distribution, reproduction and adaptation in any medium and for any purpose provided that it is properly attributed. For attribution, the original author(s), title, publication source (PeerJ Computer Science) and either DOI or URL of the article must be cited.
License URL: https://creativecommons.org/licenses/by/4.0/

Keywords: Topic model, Sparse data, Short text, Data dependency, Social media

Funding: The authors received no funding for this work.

==============================
The application of topic modeling to short texts is beset by challenges such as data sparsity and an absence of contextual information. Traditional research methods tend to prioritise high-attention and popular topics, frequently overlooking the identification of emerging topics. Consequently, subjects of a minor scale are prone to being overlooked during the topic identification process. Furthermore, in the context of topic modelling, information that varies in terms of the attention it receives is not treated equally. In order to address the aforementioned issues, a fairness-oriented topic discovery approach (MixTM-G) is proposed. This approach has been designed to facilitate the discovery of topics with different levels of attention. The proposed methodology involves the integration of normalized pointwise mutual information (NPMI) within a graph model to analyse text data. This approach leverages the correlation between data points to assess the semantic relationships between words, thus addressing the limitations posed by sparse data. The employment of graph algorithms facilitates the identification of semantically related clusters within the document graph, thereby enhancing the semantic associations between sparse data. Finally, a mixed topic modeling approach (MixTM), based on bi-grams and tri-grams combinations, is proposed to further improve topic discovery by strengthening the contextual relationships between words. The experimental results demonstrate the efficacy of the proposed method in topic modelling. In comparison to conventional methods, the proposed approach exhibits superior performance in detecting small-scale topics under equivalent conditions.

Introduction

In the context of online platforms, user-generated data, manifesting in the form of brief textual communications such as tweets, comments and posts, has emerged as a pivotal resource for data mining and knowledge discovery (Laureate, Buntine & Linger, 2023). This assertion is substantiated by its applications in domains including knowledge recommendation and network management. Nevertheless, the data is often brief and limited in information, and its volume can vary significantly across different topics. In particular, data with potential value is often smaller in volume compared to trending topics. This variability necessitates the fair modelling of both mature topics and those with sparse data within a noisy, mixed-data environment. In the domain of knowledge discovery, the timely identification and rapid detection of such valuable information is of critical importance. The ability to consistently model topics of varying scale, or to map topics at different stages of development onto a common space, would enhance the detection of new knowledge and improve the prediction of knowledge evolution trends. Therefore, we focus on fair modelling of short text data.

The objective of topic modeling techniques is to automatically identify and extract latent topics from text data. Nonetheless, the majority of conventional topic modelling techniques, including Latent Dirichlet Allocation (LDA) (Blei, Ng & Jordan, 2001), generally depend on substantial text data sets to produce consistent and significant topics (Sia & Duh, 2021). In summary, issues such as data sparsity and an absence of sufficient contextual information result in a considerable reduction in the effectiveness of existing methods, thereby limiting their applicability in a wide range of real-world scenarios. Topics that attract significant attention frequently pertain to large-scale datasets; hence, traditional topic models are effective in identifying latent topics. The substantial volume of data enables the model to statistically extract more accurate and significant topic features, thereby facilitating effective topic modelling. However, emerging topics frequently encounter the challenge of limited datasets, which results in the suboptimal performance of traditional topic models in these domains. In the absence of sufficient relevant data, the model may encounter difficulties in identifying the underlying structure of emerging topics, and may even fail to effectively distinguish between emerging and traditional topics.

In large-scale environments, popular topics tend to possess significantly larger data volumes in comparison to those that are nascent or in the early stages of popularity. In order to ascertain all topics with an imbalanced data volume size, we propose a topic discovery method in order to achieve fair modelling across datasets of different scales. This method employs a graph structure to represent documents and subsequently utilises topic modelling techniques to derive latent topic representations from the documents. This approach entails the transfer of reliance from the scale of the data to the correlation of the data, thereby reducing the impact of scale differences. Specifically, normalized pointwise mutual information (NPMI) is introduced to represent the text data as a graph, with word correlations being utilised to alleviate the effects of data sparsity. The graph model has been demonstrated to effectively capture both lexical associations and semantic relationships within the text, thereby providing more reliable information for subsequent topic modelling.

Furthermore, we employ graph algorithms to identify cohesive semantic clusters within the document graph. These semantic clusters demonstrate high levels of consistency and interrelation. The utilisation of these semantic clusters has been demonstrated to facilitate a more lucid comprehension and precise portrayal of subjects within concise textual compositions. Finally, a novel modelling technique, MixTM, is proposed for extracting latent topic structures within semantic sets. This method enhances conventional models by transforming the document collection into a mixed word pair set, thereby expanding the contextual range between words. This approach has been demonstrated to alleviate data sparsity and reduce the impact of small-scale data on topic discovery. The primary contributions of this article are as follows:

A novel graph-based methodology is proposed for the discovery of topics, which effectively mitigates the issue of data scale differences between topics. This enables fair topic modelling across datasets of varying scales, and is particularly effective in the discovery of topics within small-scale data.

A mixed topic modeling approach based on bi-grams and tri-grams is proposed, which enhances the contextual relationships between words by leveraging the constraints between them.

The remainder of the article is organised as follows: In “Related Work”, the existing literature on graph-based models and topic modelling is reviewed. In “Topic Discovering Methods”, a detailed description of the MixTM-G method is provided. In “Results”, the quality of the topics generated by the model and its ability to discover topics from both objective and subjective perspectives is evaluated. Finally, in “Conclusions”, the conclusions of the article are presented.

Related work

Graph-based representation of text

In the graphical representation of text, the graph structure effectively captures the complex relationships within a document (Yao, Mao & Luo, 2019). The nodes and edges in the graph represent multi-level associations between words and documents. Depending on the specific task, the method of graph construction can be adapted. For example, Wu, Xu & Zheng (2025), Cheng et al. (2025) and Hua et al. (2024) explores these complex relationships by modelling documents as heterogeneous graphs consisting of word nodes and document nodes. The edge weights in these graphs are derived from two components: the pointwise mutual information (PMI) between words and the term frequency-inverse document frequency (TFIDF) between words and documents. By applying convolutional operations to the document graph, global word information can be effectively integrated, thereby improving model performance. Song et al. (2025) effectively captures the potential associations between documents and words by introducing edges between original short texts, edges between short texts and their associated long texts, and edges between semantically similar words, based on the document-word dichotomous graph framework.

Typically, word co-occurrence information is obtained by calculating the frequency with which words occur together. However, in short texts, the frequency of word occurrences is often low, making it difficult to obtain reliable co-occurrence statistics. PMI measures the true association between two words by comparing their joint probability with the product of their individual probabilities, thus highlighting significant semantic relationships between word pairs. We want to achieve fair modelling of data at different scales. PMI can be used to transform the text into a graph model, replacing the data dependencies typical of traditional topic models with word correlations. However, PMI is prone to extreme value problems (Bouma, 2009), which can undermine its stability. To address this, the model uses NPMI to compute word associations, thereby increasing the stability of the association measure.

Topic modeling for short texts

Short text topic modelling aims to automatically identify the underlying topics in short texts. While traditional topic modelling methods such as LDA perform well when applied to longer documents, they face significant challenges when applied to short texts. Some studies have used word embedding-based methods (Kaleem et al., 2024; Kinariwala & Deshmukh, 2023; Liu et al., 2022; Rashid et al., 2023; Reuter et al., 2024; Uddin et al., 2024), such as pre-trained models like bidirectional encoder representations from transformers (BERT) (Kenton & Toutanova, 2019), generative pretrained transformer (GPT) (Brown, 2020), which convert words and texts into vector representations. Some other studies introduce incremental models with different priors to capture richer semantic relations; Zhang et al. (2025) introduces an integrated Gaussian and logistic coding network augmentation network in the neural inference network part to improve the performance and interpretability of topic extraction; Koochemeshkian & Bouguila (2024) combines BERTopic derived embeddings with multi-granularity clustered topic modelling (MGCTM) and introduces a Generalised delicacy distribution and a Beta-Liouville distribution as priors to improve the expressiveness of the MGCTM method; Wang et al. (2024) proposes the prompted topic model (SET) based on the Wasserstein Autoencoder to capture the semantic knowledge of words; Qiu et al. (2024) the Prompted Topic Model (PTM) for topic modelling using cue learning, which bypasses the structural constraints of LDA and variational autoencoder (VAE), thus overcoming the shortcomings of traditional topic models. These models map words into continuous spaces, capturing richer semantic information.

Some studies address the issue of short text sparsity by introducing different models for detecting emerging topics in short texts. Shi, Du & Kou (2020) uses the “spike and slab” method to distinguish between emerging and ordinary topics and to track emerging events. Zhu et al. (2022) combines recurrent neural networks (RNNs) with spatio-temporal information to mitigate data sparsity. Asgari-Chenaghlu et al. (2021) uses a combination of transformers and graph mining techniques to improve topic detection in multimodal data. While these methods perform well on large datasets, they are highly dependent on data size when applied to smaller datasets. Limited data may prevent the model from fully capturing semantic relationships or detecting emerging topics, thus affecting detection performance. In this article, we replace the reliance on data size in traditional topic modelling with data correlation, allowing for more robust and fair topic modelling across datasets of different scale sizes.

Topic discovering methods

In order to ensure fair topic discovery across different data scales, a novel topic discovery approach is proposed. This is called MixTM-G and consists of two key components: graph model construction and topic modelling. The graph model is employed to mask the discrepancies in the amount of related data between entities at different scales, while the bi-grams and tri-grams mixed approach proposed in topic modelling is used to enhance the contextual dependencies in short-text data. As illustrated in Fig. 1, the overview of structures is presented. The subsequent discussion will elaborate on these two aspects.

Figure 1 The overview of our method.

The document is converted into a graph model by computing the NPMI of word pairs within each window. In this graph model, the nodes correspond to words and the weights of the edges correspond to the NPMI values. The construction of the strong semantic set is then achieved through the utilisation of graph algorithms. The topic modelling is performed using MixTM, a word-pair mixing approach, and finally the topics of the small-scale data are discovered.

Graph model construction

In order to account for differences in data scale across topics, reliance on data volume is replaced by data correlation. Specifically, NPMI is introduced to measure the mutual relationship between words. The formula is as follows:

(1) NPMI=PMI(x,y)−log⁡p(x,y)PMI=log⁡p(x,y)p(x)p(y)

where p(x,y) represents the probability of co-occurrence of x and y, and p(x) represents the probability of occurrence of x.

The value of NPMI ranges from −1 to 1. A value of 1 indicates that the two words have perfect co-occurrence, i.e., they always appear together (e.g., “bat” and “ball” in a sports corpus, where they are inseparable). A value of 0 implies statistical independence, i.e., their co-occurrence is no more frequent than random chance. A value of −1 represents perfect mutual exclusion, where the words never occur together (e.g., “hot” and “cold” in a corpus where it is logically impossible for them to occur together).

In practical applications (Salle & Villavicencio, 2023; Yao, Mao & Luo, 2019), for ease of analysis, the value of NPMI usually takes the range of 0 to 1, with values closer to 1 indicating a stronger correlation between word pairs and values closer to 0 indicating a weaker or negligible correlation. This helps to mitigate the effects of data volume and shifts the focus to the relationships between word pairs. Building on this, we model the collection of documents as a graph.

Specifically, let G=(V,E) be an undirected graph, where V is the set of all words in the document collection: V=w1,w2,…,wn. The edges E represent the relationships between words, with the edge weight Oi,j=NPMI(wi,wj).

Using the NPMI metric, we can measure the mutual information between word pairs. If the data set for certain topics is smaller but more focused, the mutual information between word pairs will be higher, resulting in a higher NPMI value. Conversely, if noise words occur frequently in different topic clusters, the NPMI values between these word pairs will be lower. When several words have strong correlations, there will be several edges between them in the graph, forming a dense subgraph that reflects their close association as a topic. In this case, the document collection graph can be seen as an aggregation of strong semantic structures. Therefore, the task can be advanced by searching for complete subgraphs within the graph. A complete graph is one in which every node is directly connected to every other node. This high connectivity indicates a very close relationship between the nodes, which is consistent with the expectation of topic aggregation.

We use the Bron-Kerbosch algorithm (Bron & Kerbosch, 1973) to identify complete subgraphs, a classical graph algorithm designed to find cliques in an undirected graph. This algorithm efficiently finds all maximal cliques through a recursive process and is well suited to handle undirected graphs of different densities, whether dense or sparse. We input the mutual information constructed text graph as raw data to the algorithm to extract all maximal cliques. These cliques represent textual data sets with strong semantic associations at the semantic level.

Topic modeling

We propose a novel model, MixTM, which addresses the problem of data sparsity by considering a wider range of contextual information. This model is well suited to situations with limited data, as it captures subtle semantic relationships within the text and provides richer topic representations.

We construct documents as a set of mixed word pairs, assuming that the words in each pair belong to the same topic. For example, the short text “company department chip” can be represented as the word pair sets “company department”, “company chip”, “department chip”, “company department chip”. Let α and β be the prior parameters for the topic distribution θ and the topic-word distribution φ respectively, Dir means Dirichlet distribution, Multi represents the multinomial distribution, the text generation process of the model is as follows: 1. For each topic z:

Draw topic-word distribution φ∼Dir(β)

2. Draw topic distribution θ∼Dir(α)

3. For each term in the set of word pairs B

Draw z∼Multi(θ)

Draw term∼Multi(φz)

term=(wi,wj)orterm=(wi,wj,wk)

The graphical representation of the model is shown in Fig. 2. By constructing documents as a mixture of bi-grams and tri-grams, MixTM provides a richer contextual representation than the bi-gram model, significantly improving the model’s ability to distinguish between different topics. In addition, this combination captures more complex relationships between words, making the prediction of the next word more constrained and contextually informed. As a result, the topic modelling process becomes more accurate and robust, particularly in short or sparse texts where limited contextual information could hinder the discovery of meaningful topics.

Figure 2 Graphical representation model of MixTM.

In the document generation process described above, we need to infer the parameters: θ, φ, and the latent variable k. Given the simplicity and effectiveness of the Collapsed Gibbs Sampling method, we use it to estimate these parameters. The sampling formula for Gibbs sampling is as follows:

(2) p(kb|K−b,B,α,β)∝p(K,B|α,β)p(K−b,B−b|α,β).

The formula represents the conditional probability distribution of the latent variable k of the target word pair b, given the topics of all word pairs except the word pair b and the hyperparameters a and β of all other word pairs. The joint probability of all word pairs is denoted by p(K,B|α,β). The joint probability of all word pairs except the pair b is represented by p(K,B|α,β). The joint probability of all word pairs is calculated as follows:

(3) p(K,B|α,β)=p(K|α)p(B|K,β)=∏d=1DΔ(ndK+α)Δ(α)∏k=1KΔ(nkB+β)Δ(β).

In this context, ndk represents the frequency of each topic k in document d, and nkB represents the number of times a word pair belongs to topic k, β denotes the Beta function. The formula represents the joint generation probability of the topic structure and word distribution given the hyperparameters α and β. Combining the two Formulas (2) (3), the final sampling formula can be derived as:

For the pairs composed with two words ( wi, wj):

(4) p(kb=k|K−b,B,α,β)∝(nk,−b+α)(∑k=1Knk,−b+Kα)×(nk,−bwi+β)(nk,−bwj+β)(∑w=1Wnk,−bw+Wβ)(∑w=1Wnk,−bw+1+Wβ).

For the pairs composed with three words ( wi, wj, wg):

(5) p(kb=k|K−b,B,α,β)∝(nk,−b+α)(∑k=1Knk,−b+Kα)×(nk,−bwi+β)(nk,−bwj+β)(nk,−bwg+β)(∑w=1Wnk,−bw+Wβ)(∑w=1Wnk,−bw+1+Wβ)(∑w=1Wnk,−bw+2+Wβ)

where nk,−b denotes the number of word pairs assigned to topic k after excluding word pair b, nk,−bwi represents the number of occurrences of word wi in topic k.

The Gibbs sampling process commences with the random selection of an initial state and the subsequent allocation of initial values to each variable. Subsequently, the new values of the variables are recalculated using the sampling formulas. The process is iterated until a stable state is reached, at which point the topic distribution for each word is obtained. Formulas (6) and (7) are employed to calculate the parameters θ and φ, while nkw signifies the number of times the word is assigned to topic k.

(6) φk,w=nkw+β∑wnkw+Wβ

(7) θk=nkw+α∑w|B|+Kα.

It is not possible for the model to obtain the topic distribution directly from the learning process. In order to infer topics, it is assumed that the topic distribution of each document is equivalent to the expectation of the topic proportions of word pairs generated from the document.

(8) p(k|d)=∑bp(k|b)p(b|d)

and p(k|b) can be calculated by the Bayesian formula:

(9) p(k|b)=p(k)p(wi|k)p(wj|k)∑bp(k)p(wi|k)p(wj|k)orp(k|b)=p(k)p(wi|k)p(wj|k)p(wg|k)∑bp(k)p(wi|k)p(wj|k)p(wg|k)

where p(k)=θk,p(wi|k)=φk,wi, p(b|d) can be calculated using a simple uniform distribution:

(10) p(b|d)=nd(b)∑bnd(b)

nd(b) denotes the frequency of word-pair b in document d.

Results

Two experiments are set up: the first is the small data topic discovery experiment, in which small data are added to public datasets to evaluate the model’s ability to discover topics. The second experiment is the semantic representation quality experiment, in which qualitative metrics are utilised to evaluate the quality of the topics generated by the method.

Dataset

QA-data (Yan et al., 2013): This dataset is collected from the Baidu Zhidao website. THUCNews (http://thuctc.thunlp.org/): This is a news text classification dataset provided by the Tsinghua NLP group. 20NG (http://qwone.com/∼jason/20Newsgroups/): The 20 Newsgroups dataset is a standard dataset that includes 20 different topics, commonly used for topic modeling classification tasks.

In the preprocessing stage, for the Chinese dataset, we use the jieba tool to remove the stop words in the dataset, remove texts with length less than 2. For the English dataset, we use the nltk tool to tokenize the texts, convert characters to lower cases, remove texts with length less than 2 and filter out illegal characters. The dataset statistics are reported in Table 1.

Table 1 Dataset statistics.

Dataset	Number of texts	Vocabulary size	Labels	
QA-data	57,686	10,195	33	
THUCNews	80,006	21,052	7	
20NG	16,309	1,509	20	

Baselines

(1) BTM (Yan et al., 2013): A topic model specifically designed for short texts, which captures word co-occurrences in word pairs (biterms) to discover latent topics. (2) PYSTM (Niu, Zhang & Li, 2021): A topic model that uses the Pitman-Yor process for self-aggregation. (3) NQTM (Wu et al., 2020): A topic model that combines negative sampling and quantization techniques for short text topic modeling. (4) NSTM (Zhao et al., 2020): A neural topic model that learns document topics by minimizing the optimal transport (OT) distance between word distributions in documents. (5) ETM (Dieng, Ruiz & Blei, 2020): A topic model that combines traditional topic modeling with word embeddings. (6) BERTopic (Grootendorst, 2022): A topic model that generates document embeddings from pre-trained transducer-based language models, clusters these embeddings, and finally generates topic representations using a clas-based TF-IDF procedure. (7) TSCTM (Wu, Luu & Dong, 2022): a topic model that employ a new contrastive learning method with efficient and negative sampling strategies based on topic semantics.

Evaluation metrics

• Topic coherence: This metric is used to measure the quality of a topic by assessing the semantic coherence of the N words with the highest percentage in the distribution of generated topic words. A higher coherence score is indicative of greater topic consistency. In the experiment, the most widely used metrics Cv (Röder, Both & Hinneburg, 2015) and NPMI (Aletras & Stevenson, 2013) are employed to compute the consistency of the first N words: (11) CV(k)=1N∑i=1Nscos(νNPMI(wi),νNPMI(w1;N))

(12) νNPMI(wi)={NPMI(wi,wj)}j=1,2,…,N

(13) νNPMI(w1;N)={∑i=1NNPMI(wi,wj)}j=1,2,…,N

where scos means the cosine similarity function. For the whole corpus, we use the average Cv (1k∑k=1KCv(k)) and average NPMI (1K∑k=1KNPMI) to compute the coherence.

• Purity and NMI: In the experiment, the topic with the highest probability in the document distribution is designated as the predicted label for the document. The Purity and NMI metrics are utilised to assess the clustering efficacy of the model. A higher Purity and a larger NMI indicate better clustering performance of the model. (14) purity(Ω,C)=1n∑i=1Kmax|wi∩cj|

(15) NMI(Ω,C)=∑i,j|wi∩cj|nlog|wi||cj|n|wi∩cj|(∑i|wi|nlog|wi|n+∑j|wj|nlog|wj|n)/2.

Evaluation of the MixTM in topic discovery

Qualitative metrics are used to measure the quality of topics generated by the model. The same number of topics is set for MixTM and the comparison methods. Cv, NPMI, Purity and NMI are calculated and analyzed to objectively assess and compare the semantic quality of different algorithms.

The consistency scores of all models are shown in Tables 2, 3, 4 and Figs. 3, 4, 5. For the two short text datasets, QA-data and THUCNews, the proposed model consistency demonstrates superior performance in comparison to alternative models. In relation to Cv scores, it is evident that the attainment of maximum scores is achieved in scenarios where the number of topics is minimal. However, as the number of keywords increases, a decline in Cv scores becomes observable. This decline signifies that the hybrid word pair approach introduces irrelevant words to a certain extent, thereby causing a disruption to topic consistency. Conversely, the proposed model consistently attains a higher NPMI score. The NPMI metric is a normalized measure that serves to attenuate the impact of extreme keywords on consistency results, thereby reducing the influence of low-frequency keywords and chance associations on these results. The findings suggest that the proposed model is capable of attaining enhanced and more consistent results. For the standard length text set designated 20NG, the proposed model demonstrates a high degree of consistency overall. Among the compared methods, the NQTM and NSTM methods have higher Cv scores but lower NPMI scores. This is probably due to the presence of incidentally co-occurring word pairs in the generated subject keywords, which leads to inflated Cv scores. The BERTopic method demonstrates superior performance on the 20NG dataset, likely attributable to the capacity of the pre-trained knowledge base to accurately identify the jargon and complexity of the 20NG dataset. The extensive text contains sufficient semantic information to facilitate the model’s acquisition of enhanced semantic embedding, thereby ensuring greater consistency in topic information.

Table 2 Cv scores of QA-data.

Model	k = 50	k = 100	k = 150	
5	10	15	20	5	10	15	20	5	10	15	20	
BTM	0.63	0.54	0.49	0.45	0.43	0.48	0.52	0.54	0.37	0.46	0.54	0.58	
PYSTM	0.27	0.34	0.42	0.45	0.24	0.35	0.41	0.45	0.23	0.34	0.41	0.45	
NQTM	0.32	0.51	0.61	0.67	0.27	0.52	0.64	0.70	0.30	0.54	0.71	0.73	
NSTM	0.23	0.46	0.59	0.66	0.22	0.47	0.59	0.66	0.22	0.47	0.60	0.67	
ETM	0.23	0.54	0.67	0.74	0.22	0.54	0.67	0.74	0.51	0.39	0.39	0.41	
TSCTM	0.35	0.39	0.44	0.48	0.32	0.41	0.48	0.53	0.30	0.43	0.51	0.56	
BERTopic	0.64	0.50	0.41	0.38	0.65	0.46	0.39	0.39	0.62	0.43	0.39	0.42	
MixTM	0.69	0.57	0.49	0.45	0.67	0.50	0.41	0.36	0.66	0.49	0.40	0.36	

Table 3 Cv scores of THUCNews.

Model	k = 50	k = 100	k = 150	
5	10	15	20	5	10	15	20	5	10	15	20	
BTM	0.40	0.44	0.50	0.54	0.33	0.45	0.53	0.59	0.27	0.43	0.54	0.61	
PYSTM	0.31	0.28	0.32	0.34	0.31	0.28	0.31	0.34	0.29	0.29	0.31	0.34	
NQTM	0.27	0.58	0.68	0.76	0.27	0.52	0.63	0.69	0.28	0.51	0.62	0.69	
NSTM	0.30	0.60	0.72	0.78	0.30	0.60	0.72	0.78	0.30	0.60	0.72	0.78	
ETM	0.27	0.58	0.70	0.77	0.27	0.58	0.70	0.77	0.46	0.39	0.41	0.45	
TSCTM	0.30	0.51	0.62	0.68	0.28	0.52	0.64	0.70	0.27	0.54	0.65	0.71	
BERTopic	0.57	0.40	0.37	0.39	0.55	0.38	0.38	0.42	0.53	0.37	0.39	0.44	
MixTM	0.58	0.41	0.37	0.38	0.59	0.42	0.36	0.38	0.56	0.43	0.40	0.43	

Table 4 Cv scores of 20NG.

Model	k = 20	k = 30	k = 50	
5	10	15	20	5	10	15	20	5	10	15	20	
BTM	0.70	0.66	0.65	0.63	0.71	0.65	0.63	0.62	0.69	0.64	0.62	0.60	
PYSTM	0.66	0.62	0.64	0.69	0.62	0.59	0.61	0.66	0.60	0.55	0.57	0.62	
NQTM	0.42	0.29	0.24	0.23	0.42	0.30	0.26	0.25	0.41	0.27	0.25	0.25	
NSTM	0.64	0.55	0.52	0.48	0.63	0.54	0.51	0.47	0.62	0.53	0.48	0.46	
ETM	0.66	0.60	0.58	0.57	0.65	0.60	0.58	0.57	0.63	0.58	0.56	0.54	
TSCTM	0.76	0.71	0.68	0.65	0.71	0.64	0.61	0.58	0.68	0.60	0.56	0.52	
BERTopic	0.79	0.73	0.67	0.63	0.77	0.69	0.63	0.60	0.73	0.64	0.59	0.55	
MixTM	0.70	0.67	0.64	0.63	0.71	0.68	0.67	0.65	0.72	0.68	0.65	0.63	

Figure 3 NPMI scores on QA-data datasets.

Figure 4 NPMI scores on THUCNews datasets.

Figure 5 NPMI scores on 20NG datasets.

From the clustering results (Figs. 6, 7, 8), it has been demonstrated that the proposed approach enhances the efficacy of text clustering in comparison with preceding models. The BTM and the model under consideration employ word pairs for text modelling, with the size of the contextual range of the word being a determining factor. It has been established that the larger the contextual range of a word, the more co-occurring patterns it contains. This has a beneficial effect on the identification of topics and the clustering of documents. Our model extends word pair construction from two to three words. It is evident that the utilisation of a shared word pair construction window in our model results in the acquisition of enhanced word co-occurrence information. This superiority in data processing capacity leads to the model’s superior performance in comparison to BTM in terms of clustering efficacy. However, it is noteworthy that the PYSTM and NQTM models demonstrate suboptimal performance when applied to the 20NG dataset. This may be due to the fact that the plain text dataset contains a lot of repetitive information, and overfitting can result in the model failing to capture the differences between similar document labels, which can lead to all documents being clustered into the same category. The ETM approach may perform weakly on text due to insufficient embedding quality.

Figure 6 Clustering scores of QA-data dataset.

Figure 7 Clustering scores of THUCNews dataset.

Figure 8 Clustering scores of 20NG.

Efficiency analysis

The computational cost of the 20NG and QA-data datasets has been measured. The 20NG dataset is composed of normal-length texts, whereas the QA-data set consists of short texts. It is important to note that all models are computed in the same hardware conditions. Each model was executed for 100 iterations, and the mean of each iteration was subsequently obtained. The results have been presented in tabular form in Table 5.

Table 5 Average time of each iteration.

Models	Average time per iteration(s)/20NG	Average time per iteration(s)/QA-data	
BTM	23.51	2.14	
PYSTM	9.34	29.11	
NQTM	11.15	0.34	
NSTM	8.56	35	
ETM	1.84	29.29	
BERTopic	0.32	0.83	
TSCTM	0.71	11.35	
MixTM	68.98	3.37	

The model that has been demonstrated to demonstrate the highest computational efficiency is the BERTopic approach, in both plain text and short text. This is most likely due to the utilisation of pre-trained BERT in order to generate document embeddings, which consequently renders the subsequent clustering (UMAP+HDBSCAN) process computationally lightweight. The proposed model, due to the fact that it is reconstructing the text into a collection of mixed word pairs, takes approximately longer to train. When the length of each text is increased, the more word pairs are produced. It has been demonstrated that the mean elapsed time per iteration of the proposed model is greater for the 20NG dataset and less for training on QA-data.

Evaluation of small-scale data

In the experiment, 10 short text samples related to the topic of “Dongpo-Pork” have been added to both the QA-data and THUCNews datasets. The short-text data incorporated into the datasets includes characteristics of Dongpo-Pork, its historical origins, and everyday cooking methods. The models were trained with K = 50, 100, and 150 topics, respectively, to identify topics related to small-scale data and to display the top 15 words. The results obtained serve as evaluation metrics for the discovery of topics in small-scale data. Furthermore, the diversity metric was utilised to evaluate the quality of the generated topics.

The findings of the experiment conducted to ascertain the most effective methods of identifying topics from small-scale data are presented in Tables 6 and 7. It has been demonstrated that the quality of the semantic representation of generated topics is contingent upon the prevalence of relevant words and their elevated rankings. This suggests that the ability to identify small-scale topics is enhanced. The results indicated that NSTM and ETM were unable to identify keywords related to the “Dongpo-Pork” topic. In the NQTM and BTM models, related words appeared, but they were primarily noise words for the current topic. While the PYSTM model did identify some relevant topics, the generated topics were mixed. In contrast, the present method successfully identified relevant topic words such as “Dongpo-Pork,” “pork,” “officialdom,” “Chinese,” and “price,” and produced the most relevant words across different topics, with the highest rankings. The findings indicate that the proposed model exhibits superior performance in identifying topics from limited data sets when compared to existing methods.

Table 6 Topics selected by the Word “Dongpo-Pork” in the QA-data collections.

Model	K	Top15 words	
BTM	50	Opera, drama, dramas, culture, color, representation, makeup, art, distinction, character, stage, center, Dongpo-Pork, legacy, song	
100	Opera, drama, makeup, color, culture, Dongpo-Pork, stage, distinction, song, character, section, legacy, type, technique, image	
150	Opera, drama, dramas, culture, makeup, art, color, center, character, stage, Dongpo-Pork, substance, image, distinction, movie	
PYSTM	50	Event, crisis, nutrition, Dongpo-Pork, banana, pass, pork, cuisine, sister, textbook, milk powder, Chinese, water, citizen	
100	Shop, Dongpo-Pork, mall, pork, cuisine, counter, people, thought, floor, reason, water, conviction, seller, cake, register	
150	Fraction, Dongpo-Pork, cuisine, reality, pork, tidying, decimal, addition, rat, elephant, differ, woman, people, article, fraction	
NQTM	100	List, link, pass, maritime, medicine, strength, trademark, minister, Dongpo-Pork, practice, root, oil, hardware, institution, loan	
MixTM	50	Number, company, message, phone, address, name, proof, boss, business, record, account, station, agent, student, Dongpo-Pork	
100	Brand, model, fancy, man, white, Dongpo-Pork, school, color, student, price, pork, method, teacher, clothes, number	
150	Dongpo-Pork, pork, office, Chinese, price, type, lead, Hangzhou, adult, citizen, China, cuisine, dish, function, tradition	
MixTM-G	50	Tradition pork Dongpo-Pork price trigger soft sterilization function bright husband cuisine Hangzhou office China lead	
100	Pork tradition Dongpo-Pork price soft husband trigger function bright husband cuisine Hangzhou office China lead	
150	Pork Dongpo-Pork tradition soft sterilization function bright husband cuisine trigger price Hangzhou office China lead	

Table 7 Topics selected by the word “Dongpo-Pork” in the THUCNews collections.

Model	K	Top15 words	
BTM	50	Value index consumer rate sale core annual order retail goods college house finally business Dongpo-Pork	
100	Value rate sale person consume index consumer import core annual order retail Dongpo-Prok goods college house	
PYSTM	150	Gum pork Dongpo-Pork tax oilfield house transport matrix win machine price base pipe retail point	
MixTM	50	Earth magnet railway concept magnets people stock Xinhua impact shares Dongpo-Pork sport GuoHeng dragon price	
100	Passion mother speed jersey white number stack Dongpo-Pork advice love handsome wizards desert image Putian	
150	Pork Dongpo-Pork office price type lead home blanching adult citizen China trade dish salary tradition	
MixTM-G	50	Trust member Dongpo-Pork trigger soft color appetizer cycle soup Hangzhou price handware foresight kiss lead	
100	Dish pork Dongpo-Pork soft sterilization trigger appetizer bright cuisine Hangzhou price husband fondness China lead	
150	Pork Dongpo-Pork soft dish sterilization appetizer bright cuisine Hangzhou trigger price husband fondness China lead	

A comparison of MixTM and MixTM-G reveals that incorporating graph modelling enables the model to identify potential small-scale topics, even with a limited number of topics, and helps it stabilise at an optimal state. This is likely due to the fact that graph modelling effectively captures the relationships between words, thereby rendering the model more flexible in its handling of small-scale topics. Representing words and their interconnections as a graph structure enables the model to detect potential correlations and, even in conditions where data is sparse, extract features of small-scale topics. The graph structure facilitates the model’s capacity to manage noise and uncertainty, thereby ensuring more consistent learning outcomes across varying numbers of topics. This enhanced stability contributes to the model’s reliability in practical applications and facilitates more accurate reflection of latent topics in text. In summary, the integration of graph modelling into MixTM-G enhances its capacity to identify small-scale topics, while concurrently ensuring the model’s stability and efficacy across diverse contexts.

Conclusions

The study proposes a topic modelling method specifically designed for small-scale data in brief texts. This method constructs texts using a graph model and leverages the correlations between words to reduce reliance on data volume. The purpose of this is to form strong semantic clusters. The experimental results demonstrate that this method outperforms existing comparison algorithms in the topic discovery of small-scale data. Furthermore, given that short-text data in practical applications frequently manifests as data streams, future research could extend dynamic topic modelling, providing valuable references for practical applications in related fields.

Supplemental Information

Supplemental Information 1 Code and datasets.

Additional Information and Declarations

Competing Interests

The authors declare that they have no competing interests.

Author Contributions

Chuangying Zhu conceived and designed the experiments, analyzed the data, authored or reviewed drafts of the article, and approved the final draft.

Yongyu Liang conceived and designed the experiments, performed the experiments, analyzed the data, performed the computation work, prepared figures and/or tables, authored or reviewed drafts of the article, and approved the final draft.

Xinyuan Liang conceived and designed the experiments, performed the computation work, prepared figures and/or tables, and approved the final draft.

Limiao Zhong performed the computation work, prepared figures and/or tables, and approved the final draft.

Fei Xie performed the computation work, authored or reviewed drafts of the article, and approved the final draft.

Data Availability

The following information was supplied regarding data availability:

The code and the data are available in the Supplemental File.

The QA-data dataset is available at GitHub: https://github.com/xiaohuiyan/BTM.git.

20NG is a standard text classification dataset, collected by Ken Lang, available at http://qwone.com/~jason/20Newsgroups.

The THUCNews dataset collected by the Natural Language Processing Laboratory of Tsinghua University is available at http://thuctc.thunlp.org.

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
