# Peer review of "Fairness modeling for topics with different scales in short texts"

_PeerJ Computer Science, doi:10.7717/peerj-cs.2936_

## Round 0.1 · original submission · Major Revisions

Please read the reviewers' comments carefully and incorporate them in the new version. Please go through the referenced papers again and review some latest relevant work.

**Language Note:** The review process has identified that the English language must be improved. PeerJ can provide language editing services - please contact us at [email protected] for pricing (be sure to provide your manuscript number and title). Alternatively, you should make your own arrangements to improve the language quality and provide details in your response letter. – PeerJ Staff

Reviewer 1 ·

Basic reporting

This paper introduces MixTM, a novel topic modeling method designed to address fairness in short-text topic discovery. By using graph-based representations (NPMI) and mixed bi/tri-gram modeling, the proposed methods improve topic detection, especially for small-scale topics often overlooked by traditional models.

Experiments on three datasets (QA-data, THUCNews, 20NG) show that MixTM and MixTM-G outperform existing baselines in topic coherence, clustering performance, and diversity.

Experimental design

1. Unlike traditional models that struggle with small-scale topics, MixTM introduces a fairness-aware framework that reduces bias towards high-attention topics. The use of graph-based modeling and mutual information (NPMI) effectively mitigates data sparsity.

2. The manuscript is overall well-organized, with clear derivations and justifications for the use of NPMI, graph clustering, and bi/tri-gram modeling.

3. The study includes multiple datasets and compares against strong baselines, ensuring a fair evaluation.

Validity of the findings

1. MixTM-G is not well explained. What is different from MixTM?

2. Misused terms. The abstract says the new model is GraphBTM, but the introduction says MixTM.

3. The motivation focuses on the small-scale topics.

4. The generative process and Gibbs sampling are similar to BTM and MTM.

5. The experiments lack some recent baselines, like TSCTM, BERTopic, and FASTopic.

6. The experiments lack results on computational cost.

7. The reference list is quite outdated. Most of them are before 2022.

Cite this review as

Reviewer 2 ·

Basic reporting

1. Professional article structure, figures, and tables. Raw data shared.
- The paper does not have a clear structure – it seems like the proposed approach is under the related work section, which is not a good layout for a paper.
- It is great that the authors provided datasets and source code, but the paper lacks important information on the dataset languages (two are in the Chinese language) and a description of the preprocessing procedure. Provided code in its turn has a very short readme and almost no comments. It does not contain any requirements.txt or poetry.toml to safely install the dependencies. I tried to run “runExample.sh” which is stated to be a toy example, and the run failed due to hardcoded paths.
- “Table 1. Cv scores on the top-N(5,10,15,20) words of baselines and MixTM” – it is not evident which data is for which dataset, or if it is an average, it should also be mentioned.
- In Figure 4, confidence intervals can be seen, but they do not have any meaning here.
- The principle that is used for bolding some words (Table 2-3) is not evident. In other words, if we suppose that it is related to “Dongpo-Pork” terms, then why are such words as “bright”, "husband", “office”, and "lead "

2. Clear and unambiguous, professional English used throughout.
- The quality of English is sufficient, though additional proofreading is recommended, as sometimes authors struggle to express their ideas clearly. For example, Lines 16-17 “To address these issues, this paper proposes a fairness-oriented topic modeling approach, GraphBTM, designed to facilitate the discovery of topics with different levels of attention.” and Lines 132-133 “we propose a novel topic modeling approach, called GraphBTM, consisting of two key components”– GraphBTM is an existing approach, so it was not proposed here or if you propose a novel approach with the same name (which is not a good idea) you should at least explain its meaning. Line 54, “specifically designed for small-scale data in the context of short texts and large-scale data environments,” is not completely revealing the setting – if we are working with a small amount of data, what do authors mean by “large-scale environments”

3. Literature references, sufficient field background/context provided.
- In the related work section, there are almost no recent papers from 2023 and 2024, so the review should be updated. As far as problems with short text processing include a lack of contextual information, more neural-based models should be analyzed as they are able to successfully work with the semantics. The paper lacks a review of evaluation metrics to prove that coherence

4. Self-contained with relevant results to hypotheses.
Unfortunately, the main hypothesis of the authors, considering the good performance of their approach on small datasets with short texts, is not solidly supported by the results. Results from different models on two datasets that contain short texts are compared only by the Topic Diversity metric, which does not provide insight into the topic quality, so at least coherence for each of the datasets should be provided to support the claims.

5. Formal results should include clear definitions of all terms and theorems, and detailed proofs.

- Formulas are weakly described a lot of explanations are missed or incorrect, for example, Line 142 “where p(x,y) denotes the probability of x and y, and p(x)denotes the probability of x.” – is not a valid NPMI formula explanation, Line 181 – “Dir” is not defined as Dirichlet, Line 148 – incorrect usage of term NPMI, topic diversity formula is also incorrect.

- Considering a deeper understanding of the mentioned terms: Line 143-146 – strange description of NPMI values as strong and “nearly completely unrelated”.

- First description of MixTM-G (that is a variation of the approach proposed by the authors) appears only in the experiment setting description (Line 237) and should be described much earlier in the approach description section.

Experimental design

- Insufficient literature review leaves a question as to why comparison with such models as BERTopic was not conducted. If authors concentrated on inference speed, it is also not explicitly stated, and no evidence is provided.

- Selected metrics are not enough to show the performance on different datasets – at least they should be shown in combination (coherence, diversity, NMI) for all the models and datasets.

- Details of data preprocessing stage are omitted as well as significant details of proposed approach, such as description of MixTM-G.

Validity of the findings

- Rationale is not clearly stated, authors talk about datasets of different scales, while conducting experiments on 2 datasets with short examples and only one dataset with relatively big emails (20 NG). They also mention "topic tracking" in the abstract, but do not divide their datasets on train and test parts to provide some evidence on working with upcoming data.

- Approach itself is weakly defined as the formulas lacks descriptions and figure 1 does not provide any insights of triplets usage as documents graph is clearly a bigram model and if the goal was to find some communities of built knowledge graph there is also no evidence supporting it.

- Conclusions state that the proposed approach is better on topic diversity, which is not a solid metric for comparison.

Cite this review as

---

## Round 0.2 · accepted · Accept

Thanks for addressing all the comments in the previous reviews. I have examined all your responses. Your current version is ready for publication.

All the best.